# Frequency Comb Fiber Generator Based on Photonic Bandgap Amplifier

Aleksei Abramov, Dmitry Korobko * and Igor Zolotovskii

S.P. Kapitsa Scientific Research Institute, Ulyanovsk State University, 42 Leo Tolstoy Street, 432970 Ulyanovsk, Russia; rafzol.14@mail.ru (I.Z.)
* Correspondence: korobkotam@rambler.ru

**Abstract:** We report on a model of a fiber frequency comb generator that develops an approach to harmonically mode-locked fiber laser design based on dissipative four-wave mixing. In our version of this approach, we assume an amplifying one-dimensional photonic crystal as a key cavity element combining the properties of an intra-cavity filter and a power amplifier. Using standard equations describing the signal transformation in the ring cavity and the output fiber cascade, we have demonstrated the possibility of the application of the proposed model as a generator of broadband frequency comb with controllable line spacing.

**Keywords:** high repetition rate harmonically mode locked fiber laser; dissipative four-wave-mixing; photonic bandgap; frequency comb generation

## 1. Introduction

Ultra-fast lasers, delivering pulses with a repetition rate in the multi-GHz range, are of great interest for many applications as broadband optical frequency comb generators. Optical clocks, metrology, optical communication and microwave photonics are among their potential applications [1–3]. Mode-locked soliton fiber lasers combining a number of important properties, such as compactness, reliability, low cost and maintaining convenient output, are the most attractive sources for frequency comb generation from the consumer point of view. However, for generating regular pulse trains at a high repetition rate, the fiber laser should be harmonically mode-locked [4–6]. Harmonic mode-locking or periodic pulse distribution within the fiber laser cavity can be established by mutual repulsion of the pulses [6–8]. However, this regime is rather unstable and often requires additional steps to reduce timing and amplitude pulse jitter [9–13]. Another approach is dissipative four-wave-mixing (D-FWM) [14,15], when the fiber cavity comprises an intra-cavity filter with a free spectral range equal to the output pulse repetition rate. The matter of the D-FWM mechanism is that only two longitudinal cavity modes suffer from a positive net gain. The other cavity modes generated through FWM of initial modes are in the range where losses are higher than the gain, and so they could acquire energy only through an efficient parametric interaction. As a result, there is phase-locking of all interacting modes, leading to the generation of harmonically mode-locked pulses with a repetition rate equal to the frequency difference between the two initial modes [16–18]. Here, we propose a new version of this solution, using an amplifying one-dimensional photonic crystal (PC) as a high-Q etalon built in the ring fiber cavity, allowing the gain to be concentrated on two longitudinal modes of the cavity.

The standard frequency comb is a composition of equidistantly spaced coherent narrow laser lines, with the central line having the maximum brightness. Laser systems generating multi-frequency combs, with several lines dominating on the spectrum top, are of particular interest for specific applications in information optics and dynamic spectroscopy [19–21]. Here, we consider the generation of a double-frequency comb, where two lines at the

edges of the photonic bandgap give rise to the broad comb spectrum corresponding to the multi-GHz pulse train. The main element of the considered configuration is a photonic bandgap amplifier based on the structure, with a periodically modulated refractive index providing distributive feedback (DFB) between the amplified laser modes. Examples of such amplifiers include semiconductor quantum dot structures or Bragg gratings inscribed in fiber doped with rare earth ions. It is known that in such periodic structures the highest gain is achieved near the edges of the photonic bandgap, enabling the laser generation on two narrow bands. A remarkable property of the proposed configuration is possible tuning of the frequency comb, due to additional phase modulation and control of the PC parameters.

Finally, the evolution of the obtained pulse train in an output cascade, consisting of an optical fiber amplifier and segment of single-mode fiber (SMF), is considered. We have shown that the proposed scheme is promising for the formation of the broadband frequency comb consisting of two sub-combs, separated by a spectral interval determined by the bandgap width. The lines of the sub-comb are equidistant, and the estimation of the single line width shows that it does not exceed 1 MHz. The transformation of the pulse train in the output fiber cascade leads to a multi-order increase in the pulse peak power and broadening of the frequency comb width, demonstrating the significant potential of the proposed configuration in applications.

## 2. Ring Oscillator on the Base of Photonic Bandgap Amplifier

The key element of the model is the ring oscillator consisting of a photonic bandgap amplifier or an amplifying PC [22,23], an electro-optic modulator (EOM), a segment of the SMF, an isolator providing unidirectional signal propagation in the cavity and an output coupler (Figure 1).

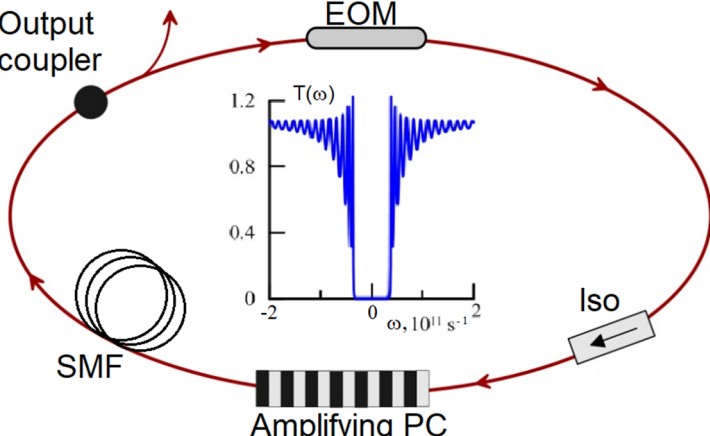

**Figure 1.** The scheme of the ring oscillator, based on a photonic bandgap amplifier. In inset: the typical transmission spectrum of the photonic bandgap amplifier.

For simplicity, we limit our consideration to a single polarization case. In this configuration of the ring oscillator, the photonic bandgap amplifier combines the functions of a high-Q intra-cavity filter and a power amplifier. Er-, Yb- or Tm-doped fiber with Bragg grating inscribed in the core or any alternative configuration of the DFB fiber laser can serve as examples of such a cavity element [24–28]. (This should not be confused with a doped fiber with a core surrounded by two-dimensional photonic bandgap cladding structure [29].) We consider this a classical one-dimensional PC with a refractive index, $n_0$, longitudinally modulated with the amplitude, $m$, so the longitudinal index distribution is given by $n(z) = n_0 + m(1 + \cos(2\pi z/\Lambda))$, where $z$ is the coordinate along the fiber. The main parameters of this structure are the coupling between the forward and backward waves, $\kappa \approx m n_0 \omega_0 / 2c$, and phase detuning, $\Delta\beta = n_0 \omega / c - \pi/\Lambda$, which are determined by the value of $m$ and the period of refractive index change, $\Lambda$. Here, $\omega_0 = 2\pi c / \lambda_0$ is the

carrier frequency. The values of parameters chosen for the description of the amplifying PC approximately corresponds to a Bragg grating inscribed in the core of Er-doped fiber with $\lambda_0 = 1560$ nm (Table 1). The assumed amplitude of the refractive index change, $m$, is quite achievable using a two-beam interferometric exposure technique or a phase mask method [30]. The amplification appearing in the pumped Er-doped fiber with inscribed Bragg grating is introduced in the standard way for a DFB laser [31]. The saturated gain factor, $g$, is averaged over the simulation window and is expressed as:

$$g = g_0 \left( 1 + \frac{1}{E_g} \int_0^{\tau_{win}} |A(t)|^2 dt \right)^{-1},$$ (1)

where $g_0$ is a small signal gain, $E_g$ is the gain saturation energy, $\tau_{win}$ is the size of the simulation window and $A$ is the signal amplitude. The gain is spectrally limited by the Gaussian filter with the parameter $\Delta\omega$ defining the gain bandwidth,

$$G(\omega) = \exp\left(-\omega^2/(2\,\Delta\omega^2)\right).$$

**Table 1.** Parameter values of the ring cavity elements used in numerical simulations.

| $\delta$ | $\beta_2$, ps$^2$km$^{-1}$ | $\beta_3$, ps$^3$km$^{-1}$ | $\gamma$, W$^{-1}$km$^{-1}$ | $g_0$, m$^{-1}$ | $\Delta\omega$, $10^{12}$ s$^{-1}$ | $m$ | PC Length, mm | $\kappa$, m$^{-1}$ | $n_0$ |
|---|---|---|---|---|---|---|---|---|---|
| 0.25 | $-20$ | 0.01 | 3 | 1.5 | $2\pi\cdot 3$ | $0.15\cdot 10^{-3}$ | 42 | 434 | 1.5 |

It is known that for the small value of $m$, the transmission of the amplifying PC can be described by the standard expression [31]:

$$T(\omega) = -\frac{S\exp(i\Delta\beta L)}{(g - i\Delta\beta)\mathrm{sh}SL - S\mathrm{ch}SL},$$ (2)

where $S^2 = \kappa^2 + (g - i\Delta\beta)^2$. An example of the amplifying PC transmission is shown in the inset in Figure 1. As one can see, the transmission spectrum in this case is marked by a pronounced band gap and its width is mainly determined by the value of $m$. For the parameter values chosen in our simulations (Table 1), the band gap width is equal to $\Omega_{PC} \approx 17.6\cdot 10^{10}$ s$^{-1}$. Specific "stopband" areas where the wave group velocity tends to zero are formed near the band gap edges. It is known that within these "stopbands" the gain is concentrated in a narrow band, less than 1 MHz wide [19–21], so it is important to match these lines with the cavity modes. Figure 2 highlights the amplifying PC's unique ability to transfer the gain into a single cavity mode. Ultimately, the gain is concentrated into two cavity modes with a frequency difference, $\Omega_{PC}$. Such an ability makes it possible to successfully implement the D-FWM mode-locking scheme in the considered configuration. Through the FWM process the energy of the "seed" modes is redistributed to higher modes and the phases of interacting modes are locked [16–18]. In the model, the coincidence between the cavity mode and PC transmission peak is ensured by a special choice of simulation window size, but in real experiments it can require the fine-tuning of the cavity length [32]. Correct adjustment of the cavity modes to the maxima of PC transmission generates a high-contrast frequency comb, with a single line width significantly less than 1 MHz. An interesting version of the model can be obtained using a PC with managed properties, where the oscillator may be subjected to fine-tuning of the band gap width and the spectral distance between two narrow gain lines [30,33].

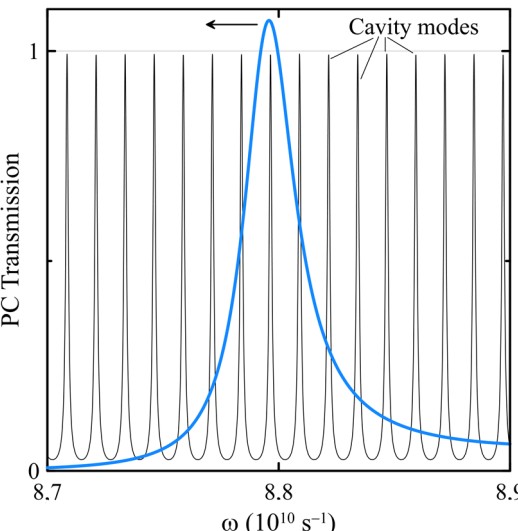

**Figure 2.** Close-up image of the area near the transmission peak of the amplifying PC. Considered PC parameter values are shown in Table 1. Arrow shows the axis for the blue curve. Modes of the 10 m cavity are also schematically shown.

The next important element of the oscillator is EOM, which is applied to control the interline spacing of the frequency comb. In this model, the EOM is described by the transmission function of the phase modulator:

$$f = \exp(i\delta \cos \Omega_m t), \tag{3}$$

where $\delta$ and $\Omega_m$ are the modulation depth and the modulation frequency, respectively [31,34]. The phase modulator, in addition to the FWM process, redistributes the energy within the cavity connecting the modes separated by modulation frequency, $\Omega_m$. The intermodal coupling efficiency is determined by the modulation depth, $\delta$. For numerical analysis, we use a GHz value of the modulation frequency, $\Omega_m$, with a small modulation depth, $\delta << \pi/2$, corresponding to real modulators [31,35]. Embedding of the phase modulator into the ring cavity allows for a significant increase in the effective depth, $\delta_{ef}$, of the signal phase modulation, $\delta_{ef} >> \delta$, and achieves the generation of the frequency sub-comb with a spacing equal to $\Omega_m$.

The light propagation in the passive SMF with length $L$ = 10 m is described by the nonlinear Schrödinger equation (NSE):

$$\frac{\partial A}{\partial z} - i\frac{\beta_2}{2}\frac{\partial^2 A}{\partial t^2} - \frac{\beta_3}{6}\frac{\partial^3 A}{\partial t^3} - i\gamma|A|^2 A = 0, \tag{4}$$

where $\beta_2$, $\beta_3$ are SMF dispersions of the second and the third orders, and $\gamma$ is the fiber Kerr nonlinearity. The 10% output coupler accumulates all the linear frequency-independent losses in the cavity. The parameter values of the cavity elements used in the numerical analysis are shown in Table 1.

The simulation time window of the size $\tau_{win} = 2^{15} \cdot 0.5578\,\text{ps} = 18.278\,\text{ns}$ was used in all the numerical experiments. The initial noise, calculated as $2^{10}$ cavity modes with random phases and Gaussian-distributed amplitudes, has been introduced into the system. After thousands of roundtrips, the oscillator comes to a stationary generation of modulated signal.

Figure 3 shows the signal spectra and intra-cavity powers obtained into the ring oscillator for different values of modulation frequency, $\Omega_m$. Since the PC transmission is strongly frequency-dependent, the cavity losses differ for varying values of $\Omega_m$. In the simulations, the parameter of gain saturation, $E_g$ energy, lies in the interval [0.75, 0,9] nJ, and its value is selected so that only two cavity modes suffer from a positive net gain and the maximum signal level remains approximately equal to 1.4–1.7 W. As one can see, at a

steady state the oscillator operates as a generator of a double-frequency comb, where the distance between the two sub-combs is equal to the band gap width, $\Omega_{PC}$. The spectral lines in the sub-combs are spaced with the frequency of the phase modulator, $\Omega_m$.

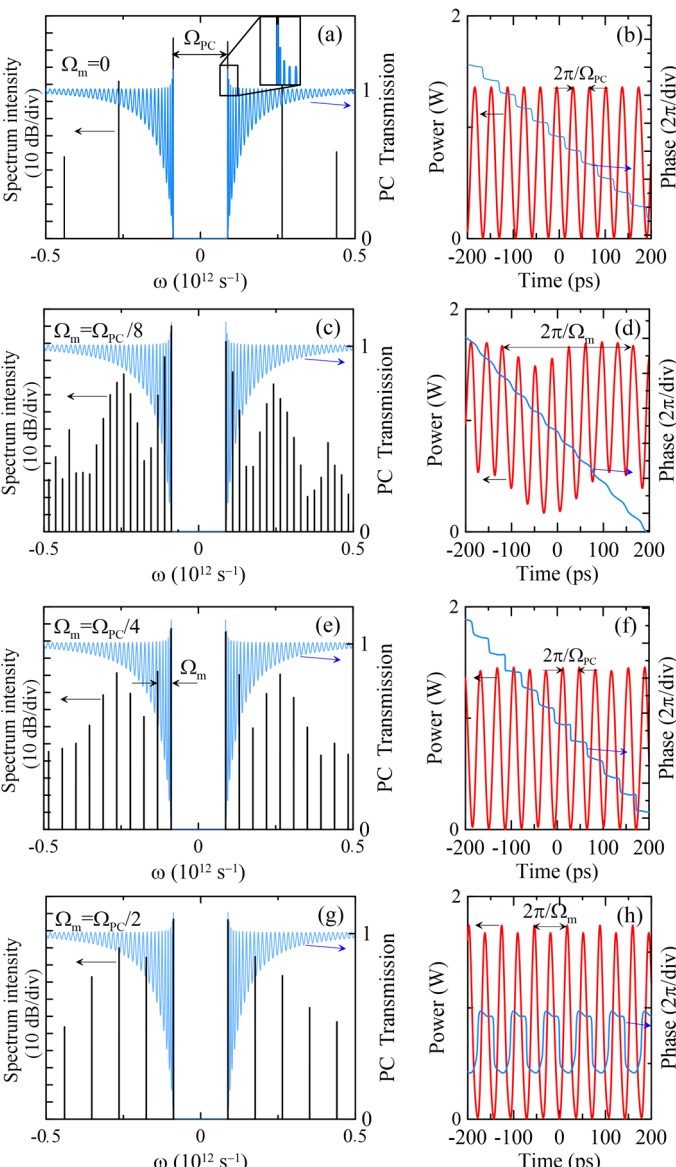

**Figure 3.** Spectra (**a,c,e,g**—black) and intra-cavity powers (**b,d,f,h**—red lines) of the signal in the ring oscillator for different modulation frequencies, $\Omega_m$. (**a,b**) Configuration without phase modulator, $E_g = 0.75$ pJ; (**c,d**) $\Omega_m = \Omega_{PC}/8$, $E_g = 0.9$ nJ; (**e,f**) $\Omega_m = \Omega_{PC}/4$, $E_g = 0.8$ nJ; (**g,h**) $\Omega_m = \Omega_{PC}/2$, $E_g = 0.8$ nJ. In (**a**) the inset shows a close-up image of the area near the transmission peak of the amplifying PC. In figures (**a,c,e,g**) the transmission spectrum of the amplifying PC is also shown (light blue). In figures (**b,d,f,h**) the blue line shows the signal phase. Arrows show the axes for corresponding curves.

Without the phase modulator (Figure 3a,b) the model is transformed into a configuration of the ring fiber laser mode-locked due to the D-FWM [16–18]. As noted above, the matter of this mechanism is that two longitudinal "seed" modes, possessing positive net gain, transfer the energy to other cavity modes through efficient parametric interaction. As a result, all interacting modes are phase-locked, leading to the generation of pulses with a repetition rate equal to the frequency difference between two initial modes. In the considered case, we observe the pulse train with the repetition rate $\Omega_{PC}/2\pi \approx 28$ GHz, corresponding to a symmetric frequency comb with strong contrast between the neighboring

lines—the intensity ratio is about 30 dB. It is worth noting that the phases of the neighbor pulses differ by a constant value, equal to $\pi$, confirming the automatic phase-locking of the pulse train.

The phase modulator introduced into the ring oscillator promotes coupling of the cavity modes, which are spaced by the modulation frequency, $\Omega_m$. Ultimately, this process leads to the excitation of new lines, forming two sub-combs divided by the band gap. The sub-comb spacing parameter is equal to the modulation frequency, $\Omega_m$. Figure 3c,e,g show examples of double-frequency combs obtained for modulation frequencies $\Omega_m = \Omega_{PC}/8, \Omega_{PC}/4, \Omega_{PC}/2$, respectively. In the time domain (Figure 3d,f,h), the combs correspond to the pulse trains with the repetition rate, $\Omega_{PC}/2\pi$, modulated with a frequency equal to $\Omega_m$. In all cases, the phase-locking of the pulse train is confirmed by the constant phase difference between the neighbor pulses with a value close to $\pi$.

### 3. Cascade Scheme of the Pulse Train Amplification and Broadband Frequency Comb Generation

In this part, we consider the cascade fiber system providing amplification and compression of the pulse train emitted by the ring oscillator. Figure 4 shows a schematic diagram of the whole system with an output fiber cascade.

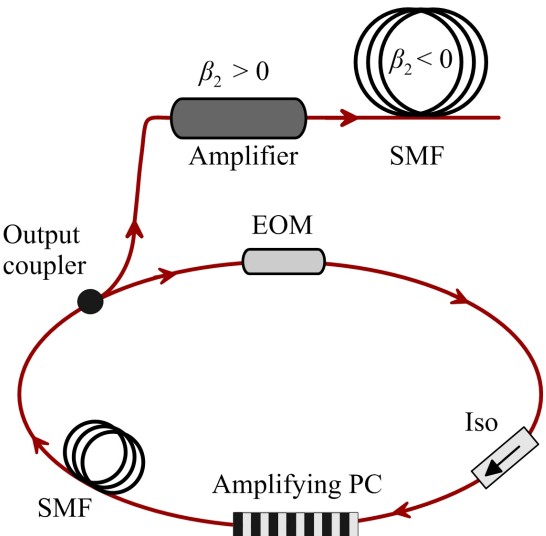

**Figure 4.** Schematic diagram of the cascade system for wide frequency comb generation.

The main elements of the fiber cascade system are a power amplifier with normal dispersion and an SMF with anomalous dispersion. For numerical simulation of the pulse train propagation in the fiber cascade, we use the standard approach on the basis of the generalized NSE, differing from Equation (4) by taking into account the spectrally limited gain and Raman response [34]:

$$\frac{\partial A}{\partial z} - i\frac{\beta_2}{2}\frac{\partial^2 A}{\partial \tau^2} - \frac{\beta_3}{6}\frac{\partial^3 A}{\partial \tau^3} + i\gamma\left(|A|^2 - \tau_R\frac{\partial |A|^2}{\partial \tau}\right)A = \frac{g_0}{2}\left(A + \frac{1}{\Delta\omega^2}\frac{\partial^2 A}{\partial t^2}\right), \qquad (5)$$

where $g_0$, $\tau_R$, $\gamma$ and $\beta_n$ are the gain, Raman response, Kerr nonlinearity and $n$-th order dispersion, respectively. For simplicity, we assume that the Raman response and third-order dispersion are the same for the fiber amplifier and for the SMF: $\tau_R \approx 3 \cdot 10^{-15}$ s and $\beta_3 = 0.01\,\text{ps}^3/\text{km}$. There is no gain in the SMF $g_0 = 0$; the dispersions and nonlinearity of the SMF are shown in Table 1. Parameter values of the amplifier are collected in Table 2.

**Table 2.** Parameter values of the fiber amplifier.

| $\beta_2$, ps$^2$/km | Nonlinearity $\gamma$, $(W \cdot km)^{-1}$ | Length, m | Gain $g_0$, $m^{-1}$ | Gain Bandwidth $\Delta\omega$, $10^{12}s^{-1}$ |
|---|---|---|---|---|
| 40 | 3 | 1 | 1.25 | $2\pi \cdot 3$ |

Assuming the output of the ring oscillator as initial conditions for the simulations of the output fiber cascade, we obtain the evolution of the signal in the time and spectral domains. This example of such an evolution in the amplifier and SMF is shown in Figure 5. As one can see, the amplifier increases the power of the signal and then, in an SMF of several hundred meters length, the pulses of the train are compressed and significant broadening of the frequency comb occurs. Because of the nonlinear parametric process of FWM and coherent addition of spectral components, the ring oscillator output is transformed into the train of short pulses with sub-picosecond duration and peak power that is orders of magnitude higher than the initial level. From Figure 5b one can see that during the propagation in the SMF, two sub-combs are merged into one frequency comb with a symmetric structure and constant interline spacing.

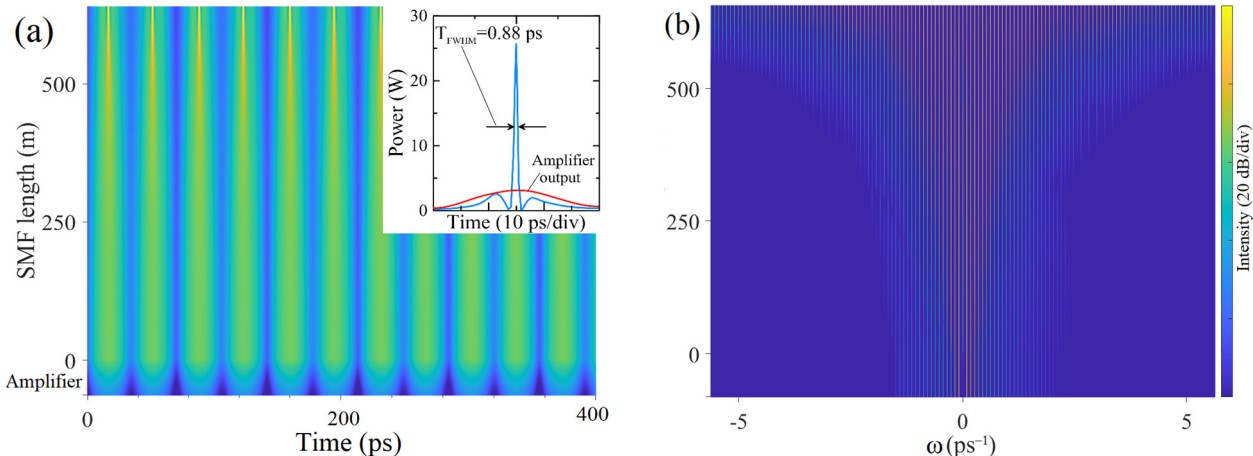

**Figure 5.** Typical evolution of the pulse train propagating through the amplifier and SMF in the time (**a**) and spectral (**b**) domains with initial conditions corresponding to Figure 2g,h. (Output of the ring oscillator with modulation frequency $\Omega_m = \Omega_{PC}/2$ and gain saturation energy $E_g = 0.8$ nJ). Inset in (**a**) shows the pulse shape after propagation in the amplifier (red) and at the output of the fiber cascade (blue line).

Figure 6 shows some features of this process for different modulation frequencies, $\Omega_m$. Initial conditions without a phase modulator (Figure 3a,b) correspond to the standard modulation instability case when the amplitude-modulated signal is transformed into the train of short pulses, with a repetition rate equal to the amplitude modulation frequency, $\Omega_{PC}$ (Figure 6a,b). The ring oscillator configuration with the phase modulator possesses two basic frequencies, $\Omega_{PC}$ and $\Omega_m$, determined by the bandgap amplifier and phase modulator, respectively. The width of the whole frequency comb is approximately equal for all the values of $\Omega_m (\sim 10^{13}$ s$^{-1}$ on $-30$ dB level, which corresponds to about 12.5 nm in the telecommunication range) and is inversely proportional to the output single-pulse duration. (From Figure 5a, one can see that $T_{FWHM} = 0.88$ ps and this value does not actually depend on the modulation frequency $\Omega_m$.) As noted above, the comb spacing parameter is equal to $\Omega_m$, but near the top of the spectral comb, additional modulation with the frequency $\Omega_{PC}$ can be observed (Figure 6c,e,g).

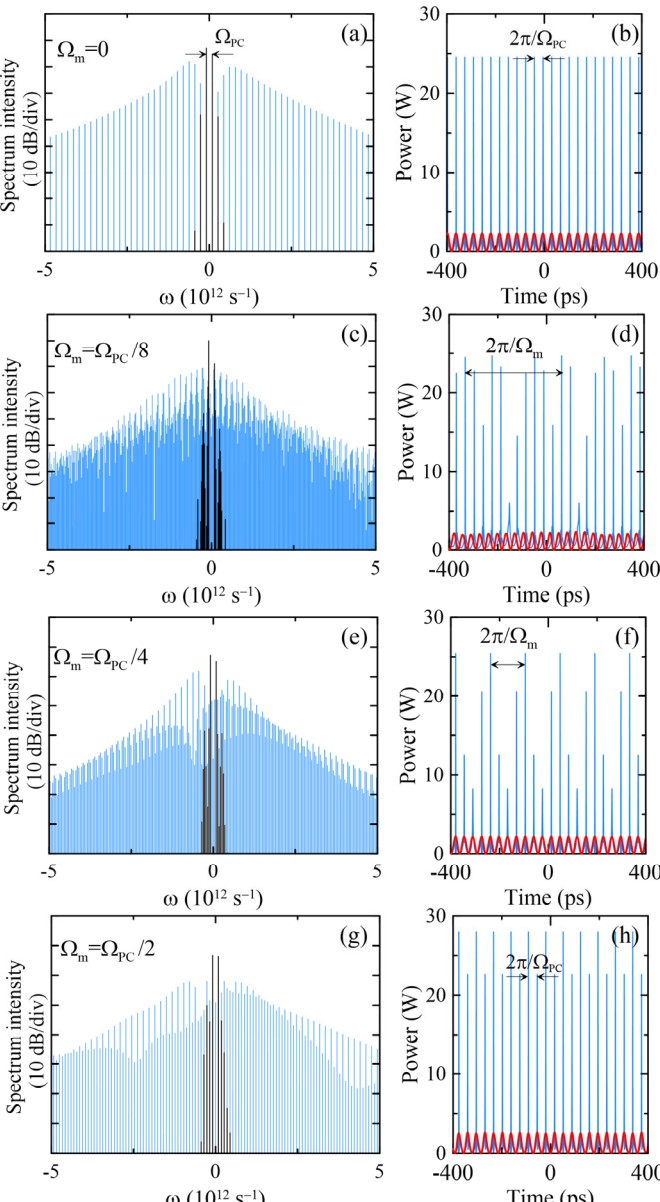

**Figure 6.** Spectra (**a**,**c**,**e**,**g**) and powers (**b**,**d**,**f**,**h**) of the signal at the output of fiber cascade (blue lines). The spectra at the output of the ring oscillator (black) and pulse trains after propagation in the amplifier (red lines) are also shown. Initial conditions for (**a**,**b**) correspond to the ring oscillator output without phase modulator (Figure 3a,b), (**c**,**d**) correspond to Figure 3c,d and so on. SMF length is 580 m (**a**,**b**), 620 m (**c**,**d**), 650 m (**e**,**f**) and 630 m (**g**,**h**).

In the time domain, the output signal of the fiber cascade is a sequence of regular pulse groups (Figure 6d,f,h) with a repetition rate of $\Omega_m/2\pi$ (~1.75, 7 and 14 GHz for (d), (f) and (h), respectively). As one can see, the pulse repetition rate within the group is determined by the bandgap width, $\Omega_{PC}/2\pi$ (~28 GHz). The peak power of pulses in the train is in the range of tens of watts, but it can be greatly increased by using an output cascade of large mode area fibers with low Kerr nonlinearity.

## 4. Discussion

We have proposed and analyzed the original model of a fiber frequency comb generator, extending the well-known approach to the development of harmonically mode-locked lasers based on D-FWM. Next, we will discuss some points regarding the experimental implementation of the proposed model. The key element of the model is amplifying a PC that

combines the functions of an intra-cavity filter and a power amplifier. Physically, examples of such structures include DFB fiber lasers [24–28], which allows the laser operation to be achieved on two "seed" lines divided by the photonic bandgap. As an example of such a structure, one can utilize the Bragg grating inscribed in the core of rare earth ion-doped fiber [24–28]. The main parameter of the PC structure is the bandgap width that determines the pulse repetition rate and maximal-frequency comb spacing. An important condition of successful comb generation is a steady overlap between the narrow transmission peak of the amplifying PC and that of the cavity mode. To solve the problem of generating stability in the experiment, it is necessary to provide isolation of the oscillator configuration from thermal and mechanical influences of the external environment, supporting the constant bandgap width and its invariable position relative to the cavity modes. In possible schemes for additional stabilization, solutions based on the optoelectronic feedback circuits can be used [36,37].

In the considered version of the configuration, the pulse repetition rate (~28 GHz) is limited by the value of the bandgap width, determined mainly by the amplitude of refractive index modulation $m$ (~$10^{-4}$). There are a number of methods allowing for the enhancement of the photosensitivity of the fiber and achieving an index change of up to $10^{-3}$ and higher (for example, due to application of hydrogen-loaded fibers [30,38]). The bandgap width of the Bragg gratings inscribed in such fibers can be significantly increased to obtain a pulse repetition rate of hundreds of GHz. Another way to develop the proposed model relies on the ability to fine-tune the bandgap width due to the local heating or mechanical deformation of fiber Bragg grating. A detailed investigation of the frequency comb variations depending on the bandgap fine-tuning may be the subject of future work.

One more question deals with the power balance in the oscillator. In the above version of the model, the losses are concentrated on the output coupler and do not exceed 10%. The gain achieved in short 42 mm Er-doped fiber with inscribed Bragg grating is enough to compensate these losses, but in real experiments, the total losses will be several times higher and an increase in the Bragg grating length could be technologically challenging. The necessary power balance in the system can be obtained with accurate selection of the pump power and length of the doped fiber, which should be longer than the Bragg grating. In a precisely balanced system, the achieved gain would compensate for the losses on the EOM, fiber splices, output coupler and provide the generation on two "seed" modes only. Finally, carefully increasing the pump power leads to the generation of a contrast frequency comb.

## 5. Conclusions

In conclusion, through applying standard equations describing the signal transformation in the ring oscillator based on the amplifying PC, we have clearly demonstrated the possibility of the generation of two frequency sub-combs with a spectral interval equal to the PC bandgap width $\Omega_{PC}$. Ultimately, after propagation through the output fiber cascade, two sub-combs are merged into the broadband frequency comb, with constant spacing equal to the frequency of the phase modulator built in the oscillator cavity.

The advantage of the proposed model is a fundamentally simple ability to control the parameters of the output signal in the frequency and time domains, achieving the generation of wide frequency combs with spacing parameters that are tunable in the multi-GHz range. We expect that the results will be of interest to a wide range of specialists in the field of radiophotonics and quantum electronics.

**Author Contributions:** Conceptualization, I.Z.; methodology, I.Z.; formal analysis, A.A.; investigation, A.A. and D.K.; data curation, A.A. and D.K.; writing—original draft preparation, A.A. and I.Z.; writing—review and editing, D.K.; supervision, D.K. All authors have read and agreed to the published version of the manuscript.

**Funding:** The research of frequency comb generation was funded by THE MINISTRY OF SCIENCE AND HIGHER EDUCATION OF THE RUSSIAN FEDERATION, grant number 075-15-2021-581. The

research of dissipative four-wave-mixing mode-locking was founded by THE RUSSIAN SCIEN-TIFIC FOUNDATION, grant number 23-79-30017. A.A. thanks the RF PRESIDENT FELLOWSHIP, fellowship number SP-4058.2021.5.

**Institutional Review Board Statement:** Not applicable.

**Informed Consent Statement:** Not applicable.

**Data Availability Statement:** Not applicable.

**Conflicts of Interest:** The authors declare no conflict of interest.

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
