# Peer review of "Frequency Comb Fiber Generator Based on Photonic Bandgap Amplifier"

_photonics, doi:10.3390/photonics10090965_

Round 1

Reviewer 1 Report

The manuscript presents a photonic bandgap amplifier based optical frequency comb generator. The bandgap and amplification property is introduced to generate comb with large spectrum range. And an EOM is applied to achieve the repetition rate adjustable. The idea is interesting and the simulation result looks good. However, there are some major revisions should be made before it can be accepted.

1.      In the abstract and the introduction section, the harmonical mode-locking is mentioned many times. However, the relationship between the harmonical mode-locking and the proposed method is not discussed. More content should be provided if the authors insist on the correlation.

2.      In the process of model establishing, the physical meaning of the parameters is not introduced enough, especially for the key parameter m. The comparison and optimization of the parameters should be presented and discussed in more details as well.

3.      How is the stability of the bandgap width? And how to revise it? If it is only in the scale of several tens of GHz, it can not be applied for wide spectrum EO-comb generation actually.

Author Response

Dear editor,

Please, find enclosed the revised version of the manuscript “Frequency comb fiber generator based on photonic bandgap amplifier”. We appreciate valuable analysis of manuscript provided by the Reviewers. We have found most of the Reviewers comments very helpful in making more clear presentation of our results. We have made corresponding revisions according to the suggestions and advices trying our best. We also add in the manuscript the discussion section and polish the English in some parts of the text.  The modifications are shown in the revised manuscript with red colored words. All the concerned issues are corrected and shown on an item-by-item basis in the following.

Best regards,              

On behalf of authors,

Dmitry Korobko

Response to reviewers' comments:

1st Reviewer.

  1. In the abstract and the introduction section, the harmonical mode-locking is mentioned many times. However, the relationship between the harmonical mode-locking and the proposed method is not discussed. More content should be provided if the authors insist on the correlation.
  2. In introduction section the mechanism of dissipative four-wave-mixing (D-FWM), providing the harmonic mode-locking is described in more detail. Additional references [14, 15] are added. Corrected text is shown below.

Another approach is dissipative four-wave-mixing (D-FWM) [14, 15] when the fiber cavity comprises an intra-cavity filter with a free spectral range equal to the output pulse repetition rate. The matter of the D-FWM mechanism is that only two longitudinal cavity modes suffer from a positive net gain. The other cavity modes generated through FWM of initial modes are in the range where losses are higher than the gain and so they could acquire energy only through an efficient parametric interaction. As a result, all interacting modes get phase-locking leading to generation of harmonically mode-locked pulses with the repetition rate equal to the frequency difference between two initial modes [16-18]. Here we propose a new version of this solution using an amplifying one-dimensional photonic crystal (PC) as a high-Q etalon built in the ring fiber cavity allowing the gain to be concentrated on two longitudinal modes of the cavity.

  1. In the process of model establishing, the physical meaning of the parameters is not introduced enough, especially for the key parameter m. The comparison and optimization of the parameters should be presented and discussed in more details as well.
  2. The description of the amplifying PC parameters has been expanded with a more detailed comment. The reference to review on fiber Bragg gratings [30] is added.

…We consider it as a classical one-dimensional PC with refractive index  longitudinally modulated with the amplitude m, so the longitudinal index distribution is given by , where z is coordinate along the fiber. The main parameters of this structure are the coupling between the forward and backward waves  and phase detuning , which are determined by the value of m and the period of refractive index change . Here is the carrier frequency. The values of parameter chosen for the description of the amplifying PC approximately correspond to a Bragg grating inscribed in the core of Er-doped fiber with nm (Table 1). The assumed amplitude of the refractive index change m is quite achievable using two-beam interferometric exposure technique or phase mask method [30]. The amplification appearing in the pumped Er-doped fiber with inscribed Bragg grating is introduced by the standard way for a DFB laser [31].

  1. How is the stability of the bandgap width? And how to revise it? If it is only in the scale of several tens of GHz, it can not be applied for wide spectrum EO-comb generation actually.
  2. Of course, these questions are very important. To answer them in detail we add the next discussion in the final part of the manuscript. New references [36-38] are also added.

Next, we will discuss some points regarding the experimental implementation of the proposed model. The key element of the model is amplifying PC that combines the functions of intra-cavity filter and power amplifier and allows to achieve the laser operation on two “seed” lines divided by the photonic bandgap. As an example of such a structure, one can utilize the Bragg grating inscribed in the core of rare earth ion doped fiber. The main parameter of the PC structure is the bandgap width that determines the pulse repetition rate and maximal frequency comb spacing. Important condition of successful comb generation is steady overlap between the narrow transmission peak of the amplifying PC and one of the cavity mode. To solve the problem of generation stability in the experiment, it is necessary to provide isolation of the oscillator configuration from thermal and mechanical influences of the external environment supporting the constant bandgap width and its invariable position relative to the cavity modes. In possible schemes for additional stabilization, solutions based on the optoelectronic feedback circuits can be used [36, 37].

In considered version of the configuration, the pulse repetition rate (~28 GHz) is limited by the value of the bandgap width determined mainly by the amplitude of refractive index modulation m (~10-4). There are a number of methods allowing to enhance the photosensitivity of the fiber and achieve an index change up to 10-3 and higher (for example, due to application of hydrogen-loaded fibers [30, 38]). The bandgap width of the Bragg gratings inscribed in such fibers can be significantly increased to obtain the pulse repetition rate of hundreds of GHz. Another way to develop the proposed model relies on the ability to fine-tune the bandgap width due to local heating or mechanical deformation of fiber Bragg grating. Detail investigation of the frequency comb variations depending on the bandgap fine-tuning may be the subject of future work.

It is worth noting that change of the bandgap width leads to variation of the maximal comb spacing but the width of the whole frequency comb (in our case, it is close to  ~ 1013 s-1 on  -30 dB level) is determined by the pulse compression in output fiber cascade.

Reviewer 2 Report

The authors proposed and analyzed the fiber frequency comb generator with amplifying PC. The possibility of application of the proposed laser configuration as a generator of broadband frequency comb with controllable line spacing was theoretically demonstrated via numerical simulation. The demonstrated results might be interesting to other groups that work in the field of frequency combs development. Thus, I believe, the manuscript may be accepted for publication in the Photonics journal.

Question:

1.      The authors propose using an amplifying PC of 42 mm length in the cavity. From the practical point of view, for some active fibers that cannot be heavily doped (e.g. Bismuth-doped fibers) this length of the active fiber might be too short to achieve enough gain. At the same time increase of the PC length might be not easy as manufacturing of long FBG could be challenging.

Can the amplifying PC be replaced by a combination of successive passive PC and an active fiber or active PC and an additional peace of active fiber in the cavity to achieve enough gain? This discussion might improve the quality of the manuscript in terms of practical applicability of the obtained theoretical results.

English language is mostly ok. Minor errors are presented but do not interfere with the understanding of the text.

Author Response

Dear editor,

Please, find enclosed the revised version of the manuscript “Frequency comb fiber generator based on photonic bandgap amplifier”. We appreciate valuable analysis of manuscript provided by the Reviewers. We have found most of the Reviewers comments very helpful in making more clear presentation of our results. We have made corresponding revisions according to the suggestions and advices trying our best. We also add in the manuscript the discussion section and polish the English in some parts of the text.  The modifications are shown in the revised manuscript with red colored words. All the concerned issues are corrected and shown on an item-by-item basis in the following.

Best regards,               

On behalf of authors,

Dmitry Korobko

Response to reviewers' comments:

2nd Reviewer

  1. The authors propose using an amplifying PC of 42 mm length in the cavity. From the practical point of view, for some active fibers that cannot be heavily doped (e.g. Bismuth-doped fibers) this length of the active fiber might be too short to achieve enough gain. At the same time increase of the PC length might be not easy as manufacturing of long FBG could be challenging.

Can the amplifying PC be replaced by a combination of successive passive PC and an active fiber or active PC and an additional peace of active fiber in the cavity to achieve enough gain? This discussion might improve the quality of the manuscript in terms of practical applicability of the obtained theoretical results.

  1. The Bragg grating should be inscribed in the doped fiber, i.e. combination of the gain and filtering ensures the generation of two “seed” modes, which eventually generate the comb spectrum. Of course, in real system the length of the doped fiber should be longer than the Bragg grating in order to compensate the losses on the EOM, fiber splices, output coupler, etc. To clear this question the next sentences are added in discussion section.

One more question deals with the power balance in the oscillator. In the above version of the model the losses are concentrated on the coupler and do not exceed 10 %. The gain achieved in short 42 mm Er-doped fiber with inscribed Bragg grating is quite enough to compensate these losses but in real experiment, the total losses will be several times higher and increase of the Bragg grating length could be technologically challenging. The necessary power balance in the system can be obtained due to accurate selection of the pump power and length of the doped fiber that should be longer than the Bragg grating. In a precisely balanced system, the achieved gain compensates the losses on the EOM, fiber splices, output coupler and provides the generation on two “seed” modes only. Finally, careful increasing of the pump power leads to the generation of contrast frequency comb.

Reviewer 3 Report

The authors report on a model of fiber frequency comb generator that develops the approach to harmonically mode-locked fiber laser design based on dissipative four-wave mixing. In this work, they have demonstrated the possibility of application of proposed model as a generator of broadband frequency comb with controllable line spacing. The work has potential to obtain the pulse trains with repetition rates up to 100 GHz and higher. The result is solid, and considering importance and novelty, I recommend the publishing of this work after several questions have been clarified.

1.      What about the function of EOM? What is the relationship between the EOM and the phase modulator described by Eq. 3?

2.      In section 2, I’m wondering about the result when the dispersion β2 of the SMF is set to be positive.

3.     Is the repetition rate of the pulse train tunable? 

The English language needs to be improved.

Author Response

Dear editor,

Please, find enclosed the revised version of the manuscript “Frequency comb fiber generator based on photonic bandgap amplifier”. We appreciate valuable analysis of manuscript provided by the Reviewers. We have found most of the Reviewers comments very helpful in making more clear presentation of our results. We have made corresponding revisions according to the suggestions and advices trying our best. We also add in the manuscript the discussion section and polish the English in some parts of the text.  The modifications are shown in the revised manuscript with red colored words. All the concerned issues are corrected and shown on an item-by-item basis in the following.

Best regards,              

On behalf of authors,

Dmitry Korobko

Response to reviewers' comments:

3rd  Reviewer

  1. What about the function of EOM? What is the relationship between the EOM and the phase modulator described by Eq. 3?
  2. In order to clear up this question the next sentences are added.

The next important element of the oscillator is EOM, which is applied to control the interline spacing of the frequency comb. In the model, the EOM is described by transmission function of the phase modulator

,        

(3)

where  and  are the modulation depth and the modulation frequency, respectively [31, 34]. The phase modulator, in addition to the FWM-process, redistributes the energy within the cavity connecting the modes separated by modulation frequency . Intermodal coupling efficiency is determined by the modulation depth .

Of course, the transmission function of the ideal phase modulator neglects the power losses on the EOM. We comment this point in discussion section.

One more question deals with the power balance in the oscillator. In the above version of the model the losses are concentrated on the coupler and do not exceed 10 %. The gain achieved in short 42 mm Er-doped fiber with inscribed Bragg grating is quite enough to compensate these losses but in real experiment, the total losses will be several times higher and increase of the Bragg grating length could be technologically challenging. The necessary power balance in the system can be obtained due to accurate selection of the pump power and length of the doped fiber that should be longer than the Bragg grating. In a precisely balanced system, the achieved gain compensates the losses on the EOM, fiber splices, output coupler and provides the generation on two “seed” modes only. Finally, careful increasing of the pump power leads to the generation of contrast frequency comb.

  1. In section 2, I’m wondering about the result when the dispersion β2 of the SMF is set to be positive.
  2. I think it is misunderstanding. The negative value of β2=-20 ps2km-1 is pointed out in Table 1. Now this table is rewritten in the next form.

, ps2km-1

, ps3km-1

,

W-1km-1

, m-1

,

1012 s-1

m

PC length, mm

, m-1

0.25

-20

0.01

3

1.5

2p

42

434

1.5

Table 1. Parameter values of the ring cavity elements used in numerical simulations.

  1.    Is the repetition rate of the pulse train tunable? 
  2. As one can see the output signal of the fiber cascade is a sequence of regular pulse groups (Fig. 6 (d, f, h)) with repetition rate . The pulse repetition rate within the group is determined by the bandgap width . The question about repetition rate tuning is answered in the next discussion.

In considered version of the configuration, the pulse repetition rate (~28 GHz) is limited by the value of the bandgap width determined mainly by the amplitude of refractive index modulation m (~10-4). There are a number of methods allowing to enhance the photosensitivity of the fiber and achieve an index change up to 10-3 and higher (for example, due to application of hydrogen-loaded fibers [30, 38]). The bandgap width of the Bragg gratings inscribed in such fibers can be significantly increased to obtain the pulse repetition rate of hundreds of GHz. Another way to develop the proposed model relies on the ability to fine-tune the bandgap width due to local heating or mechanical deformation of fiber Bragg grating. Detail investigation of the frequency comb variations depending on the bandgap fine-tuning may be the subject of future work.

Reviewer 4 Report

This manuscript presents a new method to generate the frequency comb, before it could be accepted, I think the authors should address the following problems.

1. The tile needs to be changed. "on the base of" can be written as "based on"

2. Please provide detailed specifications of the PC amplifier.

3. In Fig. 1, there is one EOM, one PC amplifier in this cavity, however, I did not see any description of the rare-earth doped fiber in this cavity. Please add more details. Another thing is that they did not provide any details about the EOM. What does it work for and how does the mode-locking realized in the cavity, is it actively mode-locking?

4. Compared with SOA, what is the advantage of a PC amplifier?

5. What is the pulse width of your pulse realized?

6. In Fig.4 they show the evolution but it is hard to distinguish the pulse width, so the authors can pick up one pulse to show in an inset, furthermore, more data should be described in the text.

7. It is recommended that the author should do some simulation about the stability and jittering of the system.

Some parts are still need to be polished.

Author Response

Dear editor,

Please, find enclosed the revised version of the manuscript “Frequency comb fiber generator based on photonic bandgap amplifier”. We appreciate valuable analysis of manuscript provided by the Reviewers. We have found most of the Reviewers comments very helpful in making more clear presentation of our results. We have made corresponding revisions according to the suggestions and advices trying our best. We also add in the manuscript the discussion section and polish the English in some parts of the text.  The modifications are shown in the revised manuscript with red colored words. All the concerned issues are corrected and shown on an item-by-item basis in the following.

Best regards,              

On behalf of authors,

Dmitry Korobko

Response to reviewers' comments:

4th Reviewer

1.The tile needs to be changed. "on the base of" can be written as "based on"

  1. Done.
  2. Please provide detailed specifications of the PC amplifier.
  3. Initially, we did not want to limit the general approach but now we connect the example of the amplifying PC with a Bragg grating inscribed in the core of Er-doped fiber. The part of the text is rewritten in the next form.

We consider it as a classical one-dimensional PC with refractive index  longitudinally modulated with the amplitude m, so the longitudinal index distribution is given by , where z is coordinate along the fiber. The main parameters of this structure are the coupling between the forward and backward waves  and phase detuning , which are determined by the value of m and the period of refractive index change . Here is the carrier frequency. The values of parameter chosen for the description of the amplifying PC approximately correspond to a Bragg grating inscribed in the core of Er-doped fiber with nm (Table 1). The assumed amplitude of the refractive index change m is quite achievable using two-beam interferometric exposure technique or phase mask method [30]. The amplification appearing in the pumped Er-doped fiber with inscribed Bragg grating is introduced by the standard way for a DFB laser [31].

  1. In Fig. 1, there is one EOM, one PC amplifier in this cavity, however, I did not see any description of the rare-earth doped fiber in this cavity. Please add more details. Another thing is that they did not provide any details about the EOM. What does it work for and how does the mode-locking realized in the cavity, is it actively mode-locking?
  2. 1) The first part of the question deals with the power balance in the oscillator. In the model we assume that gain achieved in short segment of Er-doped fiber with Bragg grating inscribed is enough to compensate 10% losses on output coupler but we agree that in real experiment the losses will be several times higher. The next text is added to comment this situation.

One more question deals with the power balance in the oscillator. In the above version of the model the losses are concentrated on the coupler and do not exceed 10 %. The gain achieved in short 42 mm Er-doped fiber with inscribed Bragg grating is quite enough to compensate these losses but in real experiment, the total losses will be several times higher and increase of the Bragg grating length could be technologically challenging. The necessary power balance in the system can be obtained due to accurate selection of the pump power and length of the doped fiber that should be longer than the Bragg grating. In a precisely balanced system, the achieved gain compensates the losses on the EOM, fiber splices, output coupler and provides the generation on two “seed” modes only. Finally, careful increasing of the pump power leads to the generation of contrast frequency comb.

2) The second part of the question deals with the EOM.  In sections 1 and 2 we tried to clear the dissipative four-wave-mixing mechanism providing the mode-locking in the oscillator. For example, the next sentences are added in intro section.

  The matter of the D-FWM mechanism is that only two longitudinal cavity modes suffer from a positive net gain. The other cavity modes generated through FWM of initial modes are in the range where losses are higher than the gain and so they could acquire energy only through an efficient parametric interaction. As a result, all interacting modes get phase-locking leading to generation of harmonically mode-locked pulses with the repetition rate equal to the frequency difference between two initial modes [16-18]. Here we propose a new version of this solution using an amplifying one-dimensional photonic crystal (PC) as a high-Q etalon built in the ring fiber cavity allowing the gain to be concentrated on two longitudinal modes of the cavity.

So in the model, the EOM works like usual phase modulator necessary to control the interline spacing of the frequency comb. The next sentences are added in the model description.

   The next important element of the oscillator is EOM, which is applied to control the interline spacing of the frequency comb. In the model, the EOM is described by transmission function of the phase modulator

,        

(3)

where  and  are the modulation depth and the modulation frequency, respectively [31, 34]. The phase modulator, in addition to the FWM-process, redistributes the energy within the cavity connecting the modes separated by modulation frequency . Intermodal coupling efficiency is determined by the modulation depth .

  1. Compared with SOA, what is the advantage of a PC amplifier?
  2. It is worth noting that the model uses the unique features of the PC amplifier, which combines an amplifier and a narrow band filter. The uniqueness of these properties is expressed in the fact that the PC amplifier can transfer the gain into two separate longitudinal modes of the cavity, thereby implementing the concept of dissipative four-wave mixing. SOA, unfortunately, does not have this selective amplification capability. In order to clear these points the fig.2 and next sentences are added.

Figure 2. Close-up image of the area near the transmission peak of the amplifying PC. Considered PC parameter values are shown in Table 1. Modes of the 10 m cavity are also schematically shown.

Fig. 2 highlights the amplifying PC's unique ability to transfer the gain into a single cavity mode. Ultimately, the gain is concentrated in two cavity modes with a frequency difference . Such ability makes it possible to implement the D-FWM mode-locking scheme in considered configuration successfully. Through FWM process, the energy of the “seed” modes is redistributed to higher modes and the phases of interacting modes get locking [16-18].

5-6. What is the pulse width of your pulse realized? In Fig.4 they show the evolution but it is hard to distinguish the pulse width, so the authors can pick up one pulse to show in an inset, furthermore, more data should be described in the text.

  1. The inset showing the pulse duration has been added. As can be seen, after compression, the output pulse duration is less than 1 ps and it does not depend on the frequency of the phase modulator.

Figure 5. Typical evolution of the pulse train propagating through the amplifier and SMF in time (a) and spectral (b) domains with initial conditions corresponding to Fig. 2 (g, h). (Output of the ring oscillator with modulation frequency  and gain saturation energy nJ). Inset in (a) shows the pulse shape after propagation in the amplifier (red) and at the output of the fiber cascade (blue line).

.. The width of the whole frequency comb is approximately equal for all the values of (~ 1013 s-1 on -30 dB level that corresponds to about 12.5 nm in telecommunication range) and it is inversely proportional to the output single pulse duration. (From Fig. 5 (a) one can see that ps and this value does not actually depend on the modulation frequency ).

  1. It is recommended that the author should do some simulation about the stability and jittering of the system.

The next discussion dealing with system stability is added.

The key element of the model is amplifying PC that combines the functions of intra-cavity filter and power amplifier and allows to achieve the laser operation on two “seed” lines divided by the photonic bandgap. As an example of such a structure, one can utilize the Bragg grating inscribed in the core of rare earth ion doped fiber. The main parameter of the PC structure is the bandgap width that determines the pulse repetition rate and maximal frequency comb spacing. Important condition of successful comb generation is steady overlap between the narrow transmission peak of the amplifying PC and one of the cavity mode. To solve the problem of generation stability in the experiment, it is necessary to provide isolation of the oscillator configuration from thermal and mechanical influences of the external environment supporting the constant bandgap width and its invariable position relative to the cavity modes. In possible schemes for additional stabilization, solutions based on the optoelectronic feedback circuits can be used [36, 37].

Round 2

Reviewer 1 Report

The manuscript has been well revised. I think it is ready for acceptance.

Author Response

We are grateful for the attention to our work. 

Reviewer 4 Report

The authors addressed most of my concerns. However, for the PC amplifier, does it really exist? I would suggest the author provide some real examples from other works.

Author Response

We are grateful to the Reviewer for the attention to our work.

In original text of the manuscript we have written that "Er, Yb or Tm doped fiber with Bragg grating inscribed in the core or any alternative configuration of the DFB fiber laser can serve as examples of such a cavity element [24-28]". References [24-28] point out the work describing the DFB fiber lasers, which can be the examples of of the PC amplifiers. In the Discussion section of the revised manuscript we additionally emphasize that "The key element of the model is amplifying PC that combines the functions of intra-cavity filter and power amplifier. Physically, examples of such structures are DFB fiber lasers [24-28], which allows to achieve the laser operation on two “seed” lines divided by the photonic bandgap. "